# Elucidating the nature of the proton radioactivity and branching ratio on the first proton emitter discovered $^{53m}$Co

The observation of a weak proton-emission branch in the decay of the 3174-keV $^{53m}$Co isomeric state marked the discovery of proton radioactivity in atomic nuclei in 1970. Here we show, based on the partial half-lives and the decay energies of the possible proton-emission branches, that the exceptionally high angular momentum barriers, $\ell_p = 9$ and $\ell_p = 7$, play a key role in hindering the proton radioactivity from $^{53m}$Co, making them very challenging to observe and calculate. Indeed, experiments had to wait decades for significant advances in accelerator facilities and multi-faceted state-of-the-art decay stations to gain full access to all observables. Combining data taken with the TASISpec decay station at the Accelerator Laboratory of the University of Jyväskylä, Finland, and the ACTAR TPC device on LISE3 at GANIL, France, we measured their branching ratios as $b_{p1} = 1.3(1)\%$ and $b_{p2} = 0.025(4)\%$. These results were compared to cutting-edge shell-model and barrier penetration calculations. This description reproduces the order of magnitude of the branching ratios and partial half-lives, despite their very small spectroscopic factors.

The observation of proton emission in the decay of the 3174-keV isomeric state in $^{53}$Co marked the discovery of proton radioactivity in 1970[1,2]. This new form of radioactive decay had already been predicted based on the estimated decay energy, the $Q_p$ value, and the calculated probability for a proton to tunnel through the Coulomb and centrifugal barriers[3,4] but it took years before it was experimentally verified. Since then, over 60 proton emitters have been discovered[5,6], and the region near doubly-magic $N = Z = 28$, $^{56}$Ni, has continued to exhibit discovery potential for exotic decay modes. For example, discrete-energy proton branches competing with $\gamma$-ray emission have been found stemming from the $10^+$ isomer in $^{54}$Ni[7,8] and from a rotational state at about 10 MeV excitation energy in $^{56}$Ni[9,10].

Until today, only one weak proton decay branch in $^{53m}$Co was known, estimated to have a branching ratio $b_p \approx 1.5\%$[11], connecting the isomeric ($19/2^-$) state with the $0^+$ ground state of $^{52}$Fe. A second branch was predicted to occur with a much weaker relative intensity of 1/250 into the first excited $2^+$ state in $^{52}$Fe[11]. A direct experimental measurement of either of the proton-emission branching ratios has not been

available for $^{53m}$Co prior to this work. In addition, this experimental input is required to allow theory to elucidate the nature of these rare decay branches having exceptionally high angular momenta, $\ell_p = 9$ and $\ell_p = 7$. Theoretically, an explanation of proton-emitting states requires a description of the wave functions of the initial and final states as well as a static or advanced time-dependent approach to the quantum tunnelling process. Model calculations typically infer values for the decay energies to derive (partial) half-lives or spectroscopic factors to be compared with experiments. The determination of the proton decay width offers a powerful means to characterize the isomeric state because of its sensitivity to the fine details of the wave function, of particular interest in $^{53m}$Co due to its peculiar structure (full alignment in the angular momentum) and its proximity to doubly-magic $^{56}$Ni.

Apart from the branching ratios, all the required information to determine the partial half-life for the proton radioactivity of $^{53m}$Co has been already measured. At the time of its discovery, the reported decay energies for proton emission from this state were $Q_p = 1560(40)$ keV[2]

e-mail: Luis.Sarmiento@nuclear.lu.se

and 1590(30) keV[11]. Since then, several measurements have improved the precision and values of 1558(8) keV[12] and 1559(7) keV[13] were determined. The most precise value thus far has been obtained in an experiment at the Accelerator Laboratory of the University of Jyväskylä, using a double Penning trap, $Q_p = 1558.8(17)$ keV[14], yielding an overall weighted average of 1558.9(16) keV.

The dominant decay mode of $^{53m}$Co is $\beta^+$ decay to its isobaric analogue state in $^{53}$Fe. This decay supports the $I^\pi = (19/2^-)$ assignment for the $^{53m}$Co state implying a full alignment of the angular momenta of one proton hole and two neutron holes in the $f_{7/2}$ shell with respect to doubly-magic $^{56}$Ni. The branching ratio $b_p \approx 1.5\%$ for proton radioactivity was estimated based on comparisons with model-dependent cross-sections for various products of the reaction $p + ^{54}$Fe. Measuring such a low proton branch is challenging because most of the decays proceed via $\beta^+$ decay, and in addition, the half-lives of the ground (240(9) ms) and the isomeric (245(10) ms) state of $^{53}$Co are nearly identical. The situation is illustrated in Fig. 1. The dominant $\beta^+$ branches have practically the same energy distribution, because both $^{53}$Co and $^{53m}$Co decay primarily into their respective isobaric analogue states in the daughter nucleus $^{53}$Fe. Secondly, any nuclear reaction aimed at producing $^{53}$Co in the high-spin isomeric state leads to a population of both $^{53}$Co and $^{53m}$Co with an a priori unknown production ratio.

In this work, we studied the proton radioactivity from $^{53m}$Co to the ground state of $^{52}$Fe and determined its absolute branching ratio, $b_{p1}$ (see Fig. 1) using a pure beam of $^{53m}$Co delivered by the JYFLTRAP Penning trap[15] to the TASISpec decay station[16]. All other ions, including $^{53}$Co in its ground state, were removed in the trapping and purification process. The relative branching ratio, $b_{p2}/b_{p1}$, of proton radioactivity from $^{53m}$Co into the first excited state and the ground state in $^{52}$Fe was measured using the ACtive TARGet and Time Projection Chamber, ACTAR TPC device[17,18] at the LISE3 separator[19] of GANIL. In this work, we have combined these state-of-the-art methodologies of ion manipulation as well as sensitive decay detection using complementary methods at world-leading facilities with cutting-edge detector setups to provide the complete description of proton radioactivity out of $^{53m}$Co, 50 years after its discovery.

## Results

One of the two experiments included in this study was carried out at the Ion-Guide Isotope Separator On-Line (IGISOL) facility[20] in the Accelerator Laboratory of the University of Jyväskylä in Finland. The isomeric-state ions, $^{53m}$Co$^+$, were produced in the fusion-evaporation reaction of protons on a $^{54}$Fe target. A quantum-state selection for the decay-spectroscopy measurements was implemented using the JYFL-TRAP double Penning trap. Purified ions were implanted into the TASISpec high-resolution charged particle-$\gamma$ coincidence set-up.

Altogether 42 h of data were collected for $^{53m}$Co, resulting in around 150000 implanted $^{53m}$Co$^+$ ions.

As seen in Fig. 2, a proton branching ratio of $b_{p1} = 1.3(1)\%$ was found to best describe the experimental energy spectrum. The best match was determined for $E_{p1,LAB} = 1537(1)$ keV, which agrees well with the experimental value derived from the $^{53m}$Co and $^{52}$Fe mass differences from the Penning-trap mass spectrometry once the dead layer of the implantation detector is accounted for using a self-consistent method between the experiment and Geant4 simulations[21].

According to Geant4 simulations, the TASISpec experiment has a sensitivity of $b_{p2} \geq 0.5\%$ for the 709-keV p2 branch. The $\beta^+$ background is too high for the identification of protons below 900 keV (see Fig. 2a), and it could not be significantly improved by requiring a coincidence with the 849-keV $\gamma$-ray connecting the $2^+$ state and the $0^+$ ground state in $^{52}$Fe (see Fig. 1). This is consistent with an upper limit of $b_{p2}/b_{p1} = 1/250$ estimated in the early experiment[11].

The ratio of the proton-emission branches of $^{53m}$Co to the first excited state and to the ground state of $^{52}$Fe was measured using the ACTAR TPC detection system at the LISE3 beam line of GANIL in another experiment. Secondary beams of $^{53}$Co ions were produced by projectile fragmentation of a $^{58}$Ni primary beam at 75 MeV/nucleon. Approximately 8% of the $^{53}$Co ions were produced in the $19/2^-$ isomeric state. During 19 h of data taking, $3.6 \times 10^6$ $^{53}$Co ions were implanted, close to 12,000 decay protons were identified, and 2167 could be further analysed. Protons leaving the active volume or directed to the cathode were disregarded in the analysis. The proton-energy spectra for two gas pressure settings of ACTAR TPC, extracted from the proton ranges in the active volume of the detector, are presented in Fig. 3. This spectrum is conditioned by a 4 s coincidence window after a $^{53}$Co implantation and a 300-ms anti-coincidence window after $^{56}$Cu and $^{57}$Zn implantation to avoid $\beta$-delayed protons from these nuclei in the data set. Both decays to the $0^+$ and to the $2^+$ states of $^{52}$Fe are visible in the spectra. For the high-pressure setting, the number of analysed p1 and p2 protons is 1563 and 45 respectively, whereas the respective numbers are 532 and 27 for the low-pressure setting. The half-life of the state was measured for the two transitions by correlating each decay event with $^{53}$Co implantations within a $\pm 4$-s time window. The resulting half-life spectra are presented in Fig. 3 and yield an average value of $T_{1/2} = 239(21)$ ms. This number agrees with the literature values, and we obtain a total average of $T_{1/2} = 245(10)$ ms.

The proton-detection efficiency, determined using a dedicated Monte-Carlo simulation, was 18.6(10)% for p1 and 25.7(3)% for p2 for the high-pressure setting and 7.7(23)% and 25.3(6)% for the low-pressure one, respectively. The resulting relative branching ratios for the $\ell_p = 7$ and the $\ell_p = 9$ decays are $b_{p2}/b_{p1} = 2.04(34)\%$ and 1.52(56)% for the different pressure settings, respectively. This yields a final ratio of $b_{p2}/b_{p1} = 1.90(29)\%$. From the above numbers, we calculate an absolute branching ratio of $b_{p2} = 0.025(4)\%$ and $b_{\beta^+/EC} = 98.67(10)\%$. The partial half-life for the proton decay of the isomer is determined to be 18.5(16) s, which yields partial half-lives of 18.8(16) s and 980(162) s for proton radioactivity to the ground and first excited state in $^{52}$Fe, respectively.

## Discussion

Experimentally measured partial half-lives can be used to probe the wave function of the isomeric state via comparison to theoretical calculations. The partial half-lives are calculated from the inverse of the decay widths multiplied by $\hbar$. The $19/2^- \to 0^+$ decay (p1 branch) involves $\ell_p = 9$ proton emission, and the $19/2^- \to 2^+$ decay (p2 branch) involves $\ell_p = 7$ proton emission (see Fig. 1). The decay rates are products of large barrier-penetration factors and tiny spectroscopic factors; in the original paper of Cerny et al.[2], a single-particle half-life for $19/2^- \to 0^+$, $\ell_p = 9$, of 60 ns was obtained from a standard barrier penetration calculation and a spectroscopic factor of $1 \times 10^{-6}$ was estimated by Peker et al.[22] in a simple model for the wave functions and residual

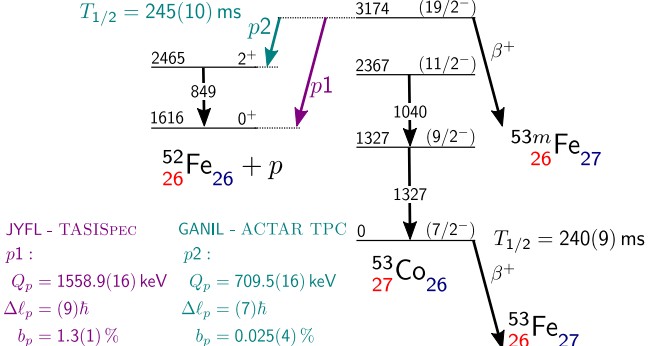

**Fig. 1 | Decay scheme of $^{53m}$Co.** The scheme is based on previous[13,47,48] and present results. Level energies are given in keV and are relative to the ground state of $^{53}$Co. The half-life of the isomer is based on the ACTAR-TPC measurement and literature[47]. The absolute proton branching ratio, $b_{p1}$, was measured with TASISpec. The relative branching ratio, $b_{p2}/b_{p1}$, was determined with ACTAR TPC.

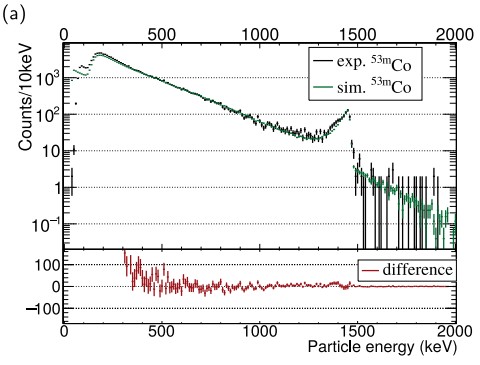

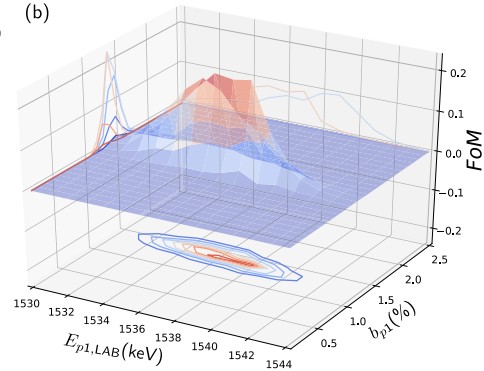

**Fig. 2 | Proton branching ratio, $b_{p1}$, determination. a** Experimental and simulated energy spectra recorded at the TASISpec implantation detector for $\beta^+$ particles and protons, the error bars represent the standard error in the number of counts. The simulated spectrum was normalized to the experimental one in the energy range 500 keV–1000 keV. The experimental data are best described when

an absolute proton branching ratio of $b_{p1} = 1.3(1)\%$ is used. The difference between the experimental and simulated spectra can be seen at the bottom. **b** Result of the Figure-of-Merit (FoM) from a minimization algorithm between experimental and simulated results.

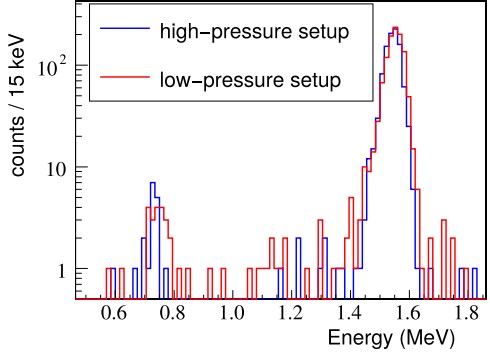

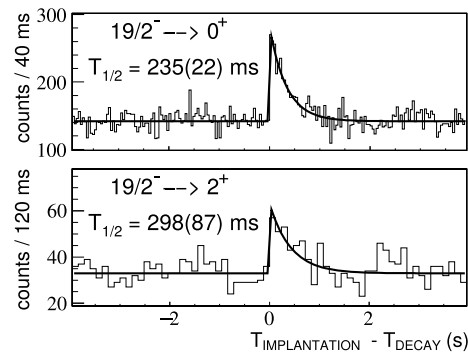

**Fig. 3 | Experimental proton energy and half-life measurement using ACTAR TPC.** Left: Proton energy spectra associated with the decay of $^{53m}$Co for the two different pressure settings (380 and 292 mbar). Right: Half-life spectra obtained by

correlating the proton emission, identified by their characteristic energy-loss profile, with $^{53m}$Co implantation for both pressure regimes. The line represents a least-squares fit of an exponential decay to the data.

interaction; none of them provided a satisfactory description. In this work we factorized the decay width into two components: a many-body nuclear structure part that gives the spectroscopic factors, $S_p$, and a potential barrier penetration part that gives the single-particle decay widths, $\Gamma_{sp}$, $\Gamma = S_p \cdot \Gamma_{sp}$.

The spectroscopic factor[23] is given by the reduced matrix element

$$S_p = \frac{|\langle \Psi(^{52}\text{Fe})_f J_f | \tilde{a}_{n,\ell,j} | \Psi(^{53}\text{Co})_i J_i \rangle|^2}{(2J_i + 1)} \quad (1)$$

where $\tilde{a}_{n,\ell,j}$ is a single-proton destruction operator, $J_i = 19/2$ and $J_f = 0$ or $J_f = 2$. The spectroscopic factors summed over all final states (f, $J_f$) gives the orbital occupation number for the orbital $(n, \ell, j)$ in the initial state (i, $J_i$). For the single-particle decay width we use a Woods-Saxon potential[24]. For a given separation energy, the high-$\ell$ single-particle wave functions are constrained to have the correct asymptotic behaviour of the decay by increasing the Woods-Saxon well depth. This calculation is combined with a Coulomb plus angular momentum barrier-penetration model with a radius parameter in agreement with results from proton scattering from a Woods-Saxon potential.

The Coulomb potential used with the Woods-Saxon calculation was obtained from a uniform charge density distribution with radius $r_c \cdot A^{1/3}$. For mass number $A = 52$, the parameter $r_c = 1.22$ fm was chosen to reproduce the experimental Coulomb displacement energy between $^{53}$Fe and $^{53}$Co of 9.07 MeV. With a diffuseness parameter of 0.67 fm, the potential radius $r_0 = 1.26$ fm was chosen to reproduce the experimental root-mean-square charge radius of $^{52}$Fe of 3.73 fm[25]. The

magnitude of the $0f_{7/2}$ proton single-particle energy of 7.15 MeV is close to the experimental proton separation energy of $^{52}$Fe, 7.38 MeV[26].

This potential was then used to calculate proton scattering from $^{52}$Fe with the code WSPOT[27]. The potential depth was adjusted to give fixed resonance energies for $\ell_p = 7$ and $\ell_p = 9$. The widths for these scattering states could numerically be obtained down to about $5 \times 10^{-11}$ MeV. For example, for $Q_p = 2.0$ MeV, $\Gamma_{sp}(\ell_p = 7) = 4.9 \times 10^{-9}$ MeV, and for $Q_p = 4.0$ MeV, $\Gamma_{sp}(\ell_p = 9) = 8.6 \times 10^{-11}$ MeV.

The single-particle proton widths were also calculated from[28]

$$\Gamma_{sp} = 2\gamma^2 P(\ell, R_c, Q_p) \quad (2)$$

with $\gamma^2 = \frac{\hbar^2 c^2}{2\mu R_c^2}$ and we obtain the Coulomb penetration, P, from Barker[29]. The channel radius, $R_c$, was chosen to match the decay widths obtained from the Woods-Saxon potential. A value of $R_c = 5.46$ fm works for both $\ell_p = 7$ and $\ell_p = 9$. With this, the barrier penetration model gives the same result as the Woods-Saxon scattering calculation over the $\Gamma_{sp}$ range from $5 \times 10^{-11}$ MeV to $1 \times 10^{-8}$ MeV to within about one percent. The barrier-penetration model can be extrapolated down to the Q values needed for proton emission of $^{53m}$Co that have single-particle decay widths of the order of $10^{-15}$ MeV. The results for $T_{1/2;sp} = \hbar \cdot \ln(2)/\Gamma_{sp}$ are given in Table 1. The uncertainties come from the uncertainty in the experimental $Q_p$ values. Relative to results for proton emission with an angular momentum of $\ell_p = 1$, the $\ell_p = 9$ decay is hindered by a factor of about $10^{12}$, and the $\ell_p = 7$ decay is hindered by a factor of about $10^9$.

**Table 1 | Results for the proton-decay calculations compared to experiment**

| $J_i^\pi$ | $J_f^\pi$ | $Q_p$ (keV) | $(n, \ell, j)$ | $T_{1/2:sp}$ ($10^{-6}$ s) | $S_p$ ($10^{-6}$) | $T_{1/2}$ [theory] (s) | $T_{1/2}$ [exp] (s) |
|---|---|---|---|---|---|---|---|
| $19/2^-$ | $0^+$ | 1558.9 (16) | (0, 9, 19/2) | 3.4 (1) | 0.062 | 55 | 18.8 (16) |
| $19/2^-$ | $2^+$ | 709.5 (16) | (0, 7, 15/2) | 58 (2) | 0.13 | 450 | 980 (162) |

The columns give the details of both the decay branches studied. Here are listed angular momenta of the orbitals involved, energies, quantum numbers, spectroscopic factors, and half-lives (single-particle, theoretical, and experimental).

For the calculation of the spectroscopic factors, we use the ($0f_{7/2}$, $0f_{5/2}$, $1p_{3/2}$, $1p_{1/2}$) (or in shorthand notation, fp) model space with the GPFX1A Hamiltonian[30]. To this, we add the configurations where one proton is moved into the high-$\ell$ orbitals that are involved in the decay. For the two-body matrix elements that connect the fp and high-$\ell$ orbitals, we used the M3Y interaction[31], and harmonic-oscillator radial wave functions with $\hbar\omega = 10$ MeV. To keep the basis dimensions tractable, the fp part of the wave function was truncated to allow for only up to one proton or one neutron to be excited from the $0f_{7/2}$ shell to one of the ($0f_{5/2}$, $1p_{3/2}$, $1p_{1/2}$) orbitals. The single-particle energies for the high-$\ell$ orbitals were placed ($\ell_p - 3$)$\hbar\omega$ above the $0f_{7/2}$ orbital. The configuration-mixing calculations were carried out with the Oxbash code[32]. The spectroscopic factors, $S_p$, obtained from these calculations and the results for the partial half-lives are given in Table 1.

The spectroscopic factors are sensitive to the model-space truncation. If we use the minimal basis of just the $0f_{7/2}$ orbital, the spectroscopic factor for $\ell_p = 7$ is increased by about a factor of 10 and the spectroscopic factor for $\ell_p = 9$ is increased by about a factor of 1.5. We have also calculated the M3Y two-body matrix elements using the Woods-Saxon radial wave functions for the artificially bound high-$\ell$ states. With this change the spectroscopic factor for $\ell_p = 7$ is increased by about a factor of two, and the spectroscopic factor for $\ell_p = 9$ is decreased by about a factor of two. The calculated results are of the same order of magnitude as the experimental values; a maximum discrepancy of a factor of two to four was found.

The decay of $^{53m}$Co, the first proton emitter ever observed, was studied in two experiments, the first at the IGISOL facility of the Accelerator Laboratory of the University of Jyväskylä and the second at the LISE3 separator of GANIL. The IGISOL data was used to determine the absolute branching ratio for proton emission of this isomeric state to the ground state of $^{52}$Fe to be 1.3(1)%. The ratio between proton emission to the first excited state relative to the ground state of $^{52}$Fe was determined with the ACTAR TPC device at GANIL yielding an absolute branching ratio to the excited state of 0.025(4)%. All decay branches of the isomer have been experimentally measured and a theoretical description has been proposed that reproduces the order of magnitude of the branching ratios and partial half-lives, despite their extremely small spectroscopic factors.

## Methods

The ions of interest for the TASISpec setup were produced using a 10 μA, 40 MeV proton beam impinging into an enriched 1.8 mg/cm$^2$ $^{54}$Fe target at the IGISOL facility[20] in the JYFL Accelerator Laboratory. The reaction products were stopped in helium gas ($P = 200$ mbar) in the IGISOL light ion guide[33] and extracted using a sextupole ion guide[34] before acceleration through a 30 kV potential. Most of the ions end up as singly charged in the helium gas. A 55° dipole magnet was used to select ions with a mass-to-charge ratio of A/q = 53. The separated ion beam was cooled and bunched in a gas-filled radio-frequency quadrupole cooler and buncher (RFQ)[35]. The ion bunches were then injected into the JYFLTRAP double Penning trap[15].

The mass-selective buffer gas-cooling technique[36] was employed to select either the isomeric-state ions ($^{53m}$Co$^+$) or the ground-state ions ($^{53}$Co$^+$) for the TASISpec post-trap decay spectroscopy setup, see Fig. 4. The ions were first held in the trap for 30 ms to cool their axial and cyclotron motions. After this initial cooling, all ions were removed from the centre of the trap by exciting them to a larger radius using a

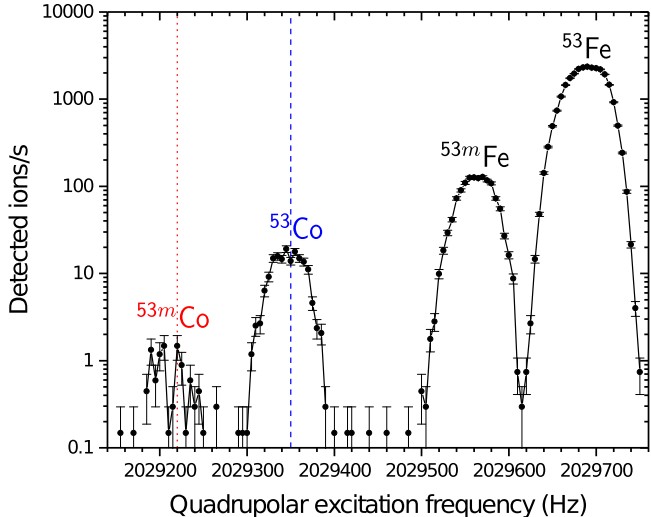

**Fig. 4 | Number of ions per second detected at the microchannel-plate detector behind JYFLTRAP.** The number is displayed as a function of the quadrupolar excitation frequency in the purification trap, the error bars represent the standard error in the number of counts. The relevant frequency region covering the ground and isomeric states of $^{53}$Co and $^{53}$Fe is shown. The red dotted line indicates one of the frequencies used for selecting the isomer $^{53m}$Co for the TASISpec measurements, the dashed blue line gives the frequency for the ground state of $^{53}$Co.

dipolar magnetron excitation for 10 ms. The mass-selective quadrupolar excitation was then applied for 80 ms at the ion's cyclotron resonance frequency. The quadrupolar excitation converts the slow magnetron motion of the ions into a much faster cyclotron motion. As a result, the ions with a cyclotron resonance frequency matching with the applied quadrupolar excitation frequency are cooled and centred in the trap due to collisions with the helium buffer gas.

To select the isomeric state $^{53m}$Co, the quadrupolar excitation frequency in the first trap was set at values of 2 029 210 Hz (first part of the data taking) and 2 029 217 Hz (second part, red dotted line in Fig. 4). For the ground state of $^{53}$Co, a frequency of 2 029 348 Hz was applied (dashed blue line in Fig. 4). The mass-selected ion bunches were extracted from the trap, accelerated to 30 keV and sent to the TASISpec setup every 141 ms.

TASISpec is composed of two subsystems, an inner silicon cube with about 80% efficiency for charged-particle detection and a surrounding array of high-purity germanium detectors for γ-ray detection. It has been extensively modelled using Geant4[37] and has been proven to allow for self-consistent cross checks of decay schemes derived from experimental spectra[38,39].

The inner subsystem covers five sides of a (~6 cm)$^3$ cube with pixelated double-sided silicon strip detectors: four 'box detectors' (16 × 16 strips, 0.97 mm thick) and one 'implantation detector' in the direction of the beam (32 × 32 strips, 0.52 mm thick). The surrounding germanium array consisted of two EUROGAM II four-fold Clover detectors[40] and one EUROBALL seven-fold Cluster detector[41]. This lowered the nominal ~40% detection efficiency at 150 keV photon energy[16] to about ~30%, which was verified using standard calibration sources of $^{133}$Ba, $^{152}$Eu, and $^{207}$Bi.

For this experiment, we performed Geant4 parameter scans of (i) the proton-emission branching ratio, $b_{p1}$, and (ii) the $E_{p1,LAB}$ value of the decay. The aim is to find the combination of parameters where experiment and simulation match best. Although the Q value for the decay was determined very precisely[14], the inclusion of the proton decay energy as a free parameter avoids any bias of the results due to any imperfection of the energy calibration of the silicon detectors. First, all simulated spectra were normalized to the experimental $^{53m}Co$ spectrum in the energy range 500 keV–1000 keV (see Fig. 2a). Second, a comparison algorithm employing the Anderson Darling test tool from ROOT[42] was used as our Figure-of-Merit to probe the match between experimental and simulated spectra for the energy range from 1000 keV to 1500 keV, which includes the proton peak p1. The result of the comparison is shown in Fig. 2b.

The second experiment was performed at the LISE3 beam line of GANIL. The $^{53}Co$ ions were produced by the fragmentation of a $^{58}Ni$ beam at 75 MeV/nucleon on a 660 μm thick beryllium target. The fragments were selected by the LISE3 spectrometer and identified using the energy loss versus time-of-flight method on an event-by-event basis. The time-of-flight was generated with the cyclotron radiofrequency and the timing signal of a fast gas detector located just before the entrance window of the ACTAR TPC indicating an ion entering the chamber.

ACTAR TPC is a gas detector that was filled for the present experiment with an Ar (95%) + $CF_4$ (5%) mixture, with 16384 read-out pads and an active volume of 25 cm × 25 cm × 20 cm working as a time-projection chamber. Each pad of the detection plane is connected to the GET electronics[43], allowing the reconstruction of the 3-dimensional tracks of the implanted ions or the emitted protons. Due to difficulties in the production process of the pad plane, some pads were grounded, creating blind zones on the detection plane. To minimize the effect of these zones on the measurement of the ratio of the proton decay branches, the measurement was performed with two pressure settings of ACTAR TPC: a high pressure ($P$ = 380 mbar) and a low pressure ($P$ = 292 mbar) setting.

Data from the LISE3 beam line and ACTAR TPC were correlated with a common dead time as well as an event-number counter and a common time stamp. The data acquisition was triggered by a signal from the fast gas detector for ion implantation, and by the pad plane multiplicity signal (with a threshold of 11 pads) for the decay events. The implantation and decay were registered as separate events by the data acquisition. When a data acquisition trigger is issued, the ACTAR TPC pad plane collected charges for 10 μs. The signal for each pad is sampled at 25 MHz after being passed through a preamplifier and a shaper. The processing of the GET electronics is already described in the literature[44,45], and more details for the present experiment can be found elsewhere[8,46].

Decay events are distinguished from implantation events by the absence of a signal in the fast gas detector and in the silicon detector upstream ACTAR TPC. Protons from the decay of the $^{53m}Co$ isomer are identified by their characteristic energy-loss profile, namely the Bragg peak, and are thus distinguished from α particles from the natural radioactivity of the ACTAR TPC material and from cosmic rays[45].

A non-negligible but undetermined fraction of the implanted $^{53}Co$ nuclei were not neutralised in the gas of ACTAR TPC and drifted towards the cathode of the detector with a drift time of the order of 3 ms, much shorter than the half-life of $^{53m}Co$. While drifting, a proportion of the ions may neutralize and will stop drifting. To obtain the correct proton-track length used to determine the proton energy in all cases, only protons directed towards the pad plane (i.e., opposite to their drift direction) with angles >20° with respect to the cathode plane were considered in the analysis to avoid "grazing" angles.

High-energy protons have a significant probability to leave the active volume of ACTAR TPC. To be considered for further analysis, the protons must be stopped within the active volume of the detector.

This is ensured for the lateral sides of the detector by the pad pattern (no signal on the external pads). To remove protons hitting the pad plane, we imposed angular cuts. Simulations allowed us to determine that p1 protons with angles <50° (high-pressure setting) or 40° (low-pressure setting) will stop in the active volume, before reaching the pad plane, independent of their starting position, i.e., the implantation height if there is no drift or higher above the pad plane for drifting ions. For p2 protons, only angles <70° are considered. This is a consequence of the pad multiplicity threshold that is not reached when the track is (close to) perpendicular to the pad surface.

To determine the impact of these restrictions and to obtain the branching ratio, $b_p$, of each proton line, simulations were performed with the Geant4 tool kit[21]. The simulated data were subjected to the above-mentioned experimental limitations to determine the proton-detection efficiencies under the same conditions. The simulations have the following ingredients: (1) an event generator for experimentally determined proton emission points from all $^{53}Co$ events and random proton-emission directions with chosen energy, (2) the simulation of the proton energy loss along its trajectory in ACTAR TPC using Geant4 with the gas pressure adjusted to reproduce the measured track length, (3) the drift of the ionization signal, with dispersion and amplification on the collection plane, and (4) the processing of the signal collected on the pads. In addition, to qualitatively determine the angular cuts necessary to ensure a proton detection efficiency independent of the height of the emission point, a fraction of the decay events was generated from the cathode plane.

Applying the same selection criteria as for the experimental data allows the determination, for each proton energy, of the global detection efficiency that combines the selection of the events as a function of the observed proton signal and the escape probability from the detection volume. As many events can be generated, the efficiency uncertainties are dominated by systematic effects due to uncertainties of the simulation parameters, rather than statistics[44].

## Data availability

The data that support the findings of this study are available from the corresponding authors L.G. Sarmiento (JYFL data) and T. Roger (ACTAR TPC data) on request (https://doi.org/10.26143/GANIL-2019-E690).

## Code availability

The WSPOT code is available at https://people.nscl.msu.edu/~brown/reaction-codes. The analysis codes used for the experimental and simulated data are available from the corresponding authors L.G. Sarmiento (JYFL data) and T. Roger (ACTAR TPC data) on request.

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

## Acknowledgements

We are grateful to the accelerator staff at GANIL and JYFL. This work was supported by the following Research Councils and Grants: European Union's Horizon 2020 Framework research and innovation programme 654002 (ENSAR2); Swedish Research Council (Vetenskapsrådet, VR 2016-3969); NSF grant PHY-1811855. Academy of Finland under the Finnish Centre of Excellence Programme 2012–2017 (Nuclear and Accelerator Based Physics Research at JYFL). The ACTAR TPC development was funded by the European Research Council under the European Union's Seventh Framework Program (FP7/2007-2013)/ERC grant agreement no 335593 and by the Conseil Régional d'Aquitaine, France (grant no 2014-1R60402—00003319). B.A.B. was supported by NSF grant PHY-2110365. A.K. acknowledges the support from the Academy of Finland under grants No. 275389, 284516 and 312544 and from the European Union's Horizon 2020 research and innovation program under grant agreement No. 771036 (ERC CoG MAIDEN). G.F.G. acknowledges the support of the Natural Sciences and Engineering Research Council of Canada (NSERC). M.C.-F., B.F.-D., and J.L.-F. acknowledge financial support from Xunta de Galicia (Spain) through grant number ED481A-2020/069, project number ED431B-2018/015, and "Centro singular de investigación de Galicia" accreditation 2019-2022, from the Spanish Research State Agency through the project PGC2018-096717-B-C22, and partial support by European Union ERDF, and the "María de Maeztu" Units of Excellence program MDM-2016-0692. B.M. is an International Research Fellow of the Japanese Society for the Promotion of Science.

## Author contributions

L.G.S. and D.R. prepared the proposal for the $^{53m}$Co experiment at JYFL with support from A.K. L.G.S., Ch.L, U.F., P.G., D.M.C., P.P., I.K., T.H., N.K., and H.S. set up the TASISpec instrument including the data acquisition, A.K., T.E., La.C., J.H., A.J., V.S.K., J.K., I.D.M., I.P., J.R., S.R.-A., and A.V. prepared the radioactive beam at IGISOL and operated the Penning trap to deliver the $^{53m}$Co beam for the JYFL experiment. L.G.S., Ch.L, D.R., C.F., U.F., N.L., M.B., and Jü.G. monitored the detectors and data acquisition. L.G.S. carried out the data analysis and interpretation of the JYFL data. B.B., D.R., and J.-C.T. prepared the proposal for the ACTAR TPC experiment, Je.G., T.R., and J.Pa. set up the instrumentation, B.B., J.-C.T., Lu.C., O.K., O.S., C.S., and J.Piot, prepared the radioactive $^{53}$Co/$^{53m}$Co beam, Je.G., T.R., B.B., D.R., H.A.-P., A.A.R., P.A., M.C.-F., Lu.C., D.M.C., B.F.-D., J.L.-F., M.G., S.G., G.F.G., B.M., A.M., J.Pa., J.Pib, J.Piot, and M.V. monitored the detector, data acquisition, and radioactive beam systems. T.R., Je.G., and A.O.M. carried out the data analysis of the ACTAR TPC experiment. B.A.B. performed the theoretical calculations. L.G.S., T.R., A.K., Je.G., B.B., D.R., and B.A.B. prepared the manuscript.

## Funding

## Competing interests

The authors declare no competing interests.

## Additional information

Luis G. Sarmiento [1] ✉, Thomas Roger [2], Jérôme Giovinazzo [3], B. Alex Brown[4], Bertram Blank [3], Dirk Rudolph [1], Anu Kankainen [5], Héctor Alvarez-Pol [6], Alex Arokia Raj[7], Pauline Ascher [3], Michael Block [8,9,10], Manuel Caamaño-Fresco [6], Lucia Caceres[2], Laetitia Canete [5,13], Daniel M. Cox[1,5], Tommi Eronen [5], Claes Fahlander[1], Beatriz Fernández-Domínguez [6], Ulrika Forsberg[1], Juan Lois-Fuentes[6], Mathias Gerbaux [3], Jürgen Gerl [8], Pavel Golubev[1], Stéphane Grévy [3], Gwen F. Grinyer [11], Tobias Habermann[8], Jani Hakala[5], Ari Jokinen[5], Omar Kamalou[2], Ivan Kojouharov[8], Veli S. Kolhinen[5,14], Jukka Koponen[5], Nikolaus Kurz [8], Nataša Lalović [1,15], Christian Lorenz[1,16], Benoit Mauss[12], Alice Mentana[7,17], Iain D. Moore [5], Aurora Ortega Moral[3], Julien Pancin[2], Philippos Papadakis [5,18], Jérôme Pibernat[3], Julien Piot [2], Ilkka Pohjalainen [5], Juuso Reinikainen[5], Sami Rinta-Antila [5], Henning Schaffner[8], Olivier Sorlin[2], Christelle Stodel[2], Jean-Charles Thomas [2], Maud Versteegen [3] & Annika Voss[5]

[1]Department of Physics, Lund University, SE-22100 Lund, Sweden. [2]Grand Accélérateur National d'Ions Lourds, CEA/DRF-CNRS/IN2P3, B.P. 55027, F-14076 Caen Cedex, France. [3]Laboratoire de Physique des Deux Infinis de Bordeaux, UMR 5797 CNRS/IN2P3—Université de Bordeaux, 19 Chemin du Solarium, F-33170 Gradignan, France. [4]Department of Physics and Astronomy and the Facility for Rare Isotope Beams, Michigan State University, East Lansing, MI 48824-1321, USA. [5]Accelerator Laboratory, Department of Physics, University of Jyväskylä, P.O. Box 35, FI-40014 Jyväskylä, Finland. [6]IGFAE - Universidade de Santiago de Compostela, E-15782 Santiago de Compostela, Spain. [7]Instituut voor Kern- en Stralingsfysica, KU Leuven, Leuven B-3001, Belgium. [8]GSI Helmholtzzentrum für Schwerionenforschung GmbH, D-64291 Darmstadt, Germany. [9]Helmholtz Institute Mainz, D-55099 Mainz, Germany. [10]Johannes Gutenberg-Universität Mainz, D-55099 Mainz, Germany. [11]Department of Physics, University of Regina, Regina, Saskatchewan S4S 0A2, Canada. [12]RIKEN Nishina Center, Wako Saitama 351–0198, Japan. [13]Present address: Department of Physics, University of Surrey, Guildford GU2 7XH, UK. [14]Present address: Cyclotron Institute, Texas A&M University, College Station, TX 77843-3366, USA. [15]Present address: Physikalisch-Technische Bundesanstalt, D-38116 Braunschweig, Germany. [16]Present address: Ingenieurgesellschaft Auto und Verkehr, D-38518 Gifhorn, Germany. [17]Present address: Physics Department, University of Pavia, I-27100 Pavia, Italy. [18]Present address: STFC Daresbury Laboratory, Daresbury, Warrington WA4 4AD, UK. ✉e-mail: Luis.Sarmiento@nuclear.lu.se

