## [Peer Review File · Nature Communications]

REVIEWER COMMENTS

Reviewer #1 (Remarks to the Author):

The paper reports on a detailed study of proton emission from the $19/2^-$ isomeric state in ^{53}Co . The experimental results come from two different experiments. In the first, conducted at the Accelerator Laboratory of the University of Jyväskylä, the absolute branching ratio for the dominating proton transition, depopulating the isomer to the ground state of ^{52}Fe , was measured. In the second, carried out at GANIL laboratory, the probability ratio of two proton transitions, a very weak branch to the first excited state in ^{52}Fe and the main one mentioned above, was determined. Both experiments are examples of advanced, modern nuclear spectroscopy. In Jyväskylä, the JYFLTRAP Penning trap was used to form a pure beam of ^{53}Co in the isomeric state, and the TASISpec array was used to detect decays by proton emission. At GANIL, the LISE3 fragment separator was used to produce and select ions on ^{53}Co , some of which (8%) were in the isomeric state, and to observe their decays by means of the ACTAR TPC gaseous detector. This technique allowed the identification of the very weak proton branch, and the measurement of its relative intensity. By combining results from the two approaches the authors established the absolute branchings for the two proton emission channels, thus completing the decay scheme of the $^{53}\text{mCo } 19/2^-$ isomer. Next, this scheme was confronted with theoretical models. Using barrier penetration calculations with a large scale shell model, it was possible to reproduce the experimental findings with a reasonably good accuracy. Overall, this work provides new data on the decay of the isomeric state and the theoretical

explanation of its decay by proton emission. This is a very nice result, representing the state-of-the-art in the low-energy nuclear physics. It certainly merits publication in a very good physics journal.

I am not sure, however, whether it meets all criteria of acceptance by Nature Communications. It does not open any new window for the further investigation of nuclear structure, neither is answers any important question in the field. The adopted experimental technique is advanced and attractive but not new. A very similar study, on the fine structure in the proton emission from the 10+ isomeric state in ^{54}Ni , was published by the same group in Nature Communications, see Ref. 7. Finally, the golden anniversary of the discovery of proton emission from ^{53m}Co , mentioned a few times in the manuscript, should not be taken as an argument in favour, as it is not scientific.

Regardless of the final decision on this manuscript, I would like to suggest some, mostly minor, corrections to the text.

1. The last sentence of the Abstract (very long!) is not grammatical, and needs rewording.

2. Line 55: the energy of the $19/2^-$ isomer is given as 3174 keV. The same value is shown in Fig.1 with the reference to the NNDC data base (Ref.19). However, the NNDC gives 3197 keV for this state. If the authors use a newer value, the proper reference should be given. Similarly, the half-life of the isomer is given by NNDC as 247(12) ms, and not 245(10) ms, as in line 96 and in Fig.1. It should be made consistent or explained.

3. Line 60: for the list of proton emitters the Ref. 5 is given.

A somewhat newer version was given by Pfutzner et al RMP 84 (2012) 567.

The most up-to-date source is the Berkeley data base:

<https://nucldata.berkeley.edu/research/betap.html>

3. Line 77: the sentence is not clear, something is missing.

4. Line 83: invoking the center-of-mass is confusing and it does not make much sense as decays in question occur at rest. Instead, it should be specified if the proton energy is meant, or the decay energy. By comparing the values in line 88 and in Fig.1, apparently the authors mean the decay energy (Q_p) but then, the Q-symbol should be used instead of $E_{\{p,CM\}}$. So please explain and make it consistent.

5. Line 344: the distinction of protons from alpha particles in the ACTAR TPC could be proven by showing examples of such events, or by providing a reference to such examples.

6. In the Summary (from line 384) it would be useful to state what is meant by "a complete understanding" of the decay under study. One could summarize here what is the theoretical picture of this decay and what is the achieved breakthrough, if any.

7. Typographical points:

- in the nuclear literature an isomer is usually denoted by placing the 'm' in superscript between the mass number and the chemical symbol (^{53m}Co). Placing the 'm' on the right side may be confusing when it appears with the charge of an ion ($m+$), like in lines 124, 128;

- the physical units, like MeV, fm, mm, should be written in regular

font, and not italic;

- line 295: a mathematical sign was rendered wrongly.

Reviewer #2 (Remarks to the Author):

The one-proton radioactivity of the $19/2^-$ isomeric state of Co-53 has been attracted attention by researchers for many years. For the transition from $19/2^-$ isomeric state of Co-53 to 0^+ ground state of Fe-52, its branching ratio was determined as about 1.5% in 1972. Although the one-proton radioactivity of Co-53m was remeasured in the past decades, the branching ratio from $19/2^-$ to the first excitation state (2^+ state) was not determined accurately because its branching ratio is so small ($<0.006\%$) that it is impossible to observe the proton radioactivity with such a small branching ratio. In this manuscript, the authors measured the branching ratios from $19/2^-$ to 0^+ and 2^+ states by the JYFL-TASISPEC and GANIL-ACTAR TPC. The two branching ratios were determined as 1.3(1)% and 0.025(4)%, respectively. Furthermore, the hindered proton-radioactivity with high angular momentum was analyzed by shell models. So this work is interesting and important for understanding the structure and decay properties of Co-53. Before publication on NC, a few minor revisions are advised to be considered by the authors:

1. In paragraph 1 of page 5, the authors expressed "...which yields partial half-lives of 18.8(16) s and 968 (160) s...". However, in the last column and the first line of Table 1, the listed partial half-life is 18.6(16) s. The data must be checked by the authors.

2. In the discussion section, the spectroscopic factor was estimated. However, the details for calculating the spectroscopic factor was not shown. At least, a few equations or formulas that are correlated to the spectroscopic factor should be given.

3. Has the spectroscopic factor been calculated by other models? What's the order of magnitude within by other models? Are the spectroscopic factors dependent on different models?

4. In paragraph 2 of page 6, the proton-decay width $\Gamma_p = W P(\rho)$. What does the "W" stand for?

Reviewer #3 (Remarks to the Author):

In the proton radioactivity a quasi-bound proton tunnels through the Coulomb and centrifugal barrier, leaving the initial state and populating states in the nucleus with one less proton. Naturally, this phenomenon occurs in proton rich nuclei

at or near the proton drip line.

The present paper deals with the proton decay of the $^{53}\text{Co}^m$ isomer, in which the proton radioactivity was first observed already 50 years ago. Now it is possible to study the process quantitatively and fully, and determine even the branching ratio of the very weakly populated $2+$ state in the final nucleus ^{52}Fe .

This is in some sense a continuation of Ref. [7] published in Nature Communications in 2021 by a subset of the coauthors of the present paper. Ref. [7] dealt with the proton decay of $^{54}\text{Ni}^m$ isomer, i.e. the nucleus with one more proton than here. Analogous devices, e.g. the ACTAR TPC detector at GANIL are used in both papers; the theoretical analyses are also similar in both papers. However, the present paper not only uses in addition the JYLTRAP Penning trap at Jyvaskyla, but deals with much weaker, thus more difficult to observe, proton decay branches. I believe that the Nature Communications journal is the right place to publish the present results.

My minor suggestion is to add, either on a separate line or in the text, the formula connection the partial half-life with the width ($T_{1/2\text{ sp}} = \hbar \ln(2) / \Gamma_{\text{sp}}$) as well as the partial half-life with the total one ($T_{1/2} = T_{1/2\text{ sp}} / C_{2S}$). That would make the understanding of Table 1 easier to an educated non-expert.

I agree with the authors that the agreement between the measured and calculated branching ratios is remarkable given the approximations used. Not only is the model space truncated, with only one nucleon allowed to move out of the $f_{7/2}$ orbit, but also the barrier penetration model is extrapolated by many orders of magnitude from the

$\Gamma_{\text{sp}} \sim 10^{-11}$ MeV to about 10^{-16} MeV (the text mentioned 10^{-15}), I believe the entries in Table 1 correspond rather to about 10^{-16}). The theoretical uncertainty might be actually larger than the factor of three mentioned. I hesitate

what to suggest but maybe instead of 'close to a factor of three' one could say

'at least a factor of three'.

I leave it to the authors whether they choose to follow my minor suggestions or not.

I recommend the publication in Nature Communications.

Replies to the reviewers

Reviewer #1 (Remarks to the Author):

The paper reports on a detailed study of proton emission from the $19/2^-$ isomeric state in ^{53}Co . The experimental results come from two different experiments. In the first, conducted at the Accelerator Laboratory of the University of Jyväskylä, the absolute branching ratio for the dominating proton transition, depopulating the isomer to the ground state of ^{52}Fe , was measured. In the second, carried out at GANIL laboratory, the probability ratio of two proton transitions, a very weak branch to the first excited state in ^{52}Fe and the main one mentioned above, was determined. Both experiments are examples of advanced, modern nuclear spectroscopy. In Jyväskylä, the JYFLTRAP Penning trap was used to form a pure beam of ^{53}Co in the isomeric state, and the TASI Spec array was used to detect decays by proton emission. At GANIL, the LISE3 fragment separator was used to produce and select ions on ^{53}Co , some of which (8%) were in the isomeric state, and to observe their decays by means of the ACTAR TPC gaseous detector. This technique allowed the identification of the very weak proton branch, and the measurement of its relative intensity. By combining results from the two approaches the authors established the absolute branchings for the two proton emission channels, thus completing the decay scheme of the ^{53m}Co $19/2^-$ isomer. Next, this scheme was confronted with theoretical models. Using barrier penetration calculations with a large scale shell model, it was possible to reproduce the experimental findings with a reasonably good accuracy. Overall, this work provides new data on the decay of the isomeric state and the theoretical explanation of its decay by proton emission. This is a very nice result, representing the state-of-the-art in the low-energy nuclear physics. It certainly merits publication in a very good physics journal. I am not sure, however, whether it meets all criteria of acceptance by Nature Communications. It does not open any new window for the further investigation of nuclear structure, neither it answers any important question in the field. The adopted experimental technique is advanced and attractive but not new. A very similar study, on the fine structure in the proton emission from the 10^+ isomeric state in ^{54}Ni , was published by the same group in Nature Communications, see Ref. 7. Finally, the golden anniversary of the discovery of proton emission from ^{53m}Co , mentioned a few times in the manuscript, should not be taken as an argument in favour, as it is not scientific.

Regardless of the final decision on this manuscript, I would like to suggest some, mostly minor, corrections to the text.

1. The last sentence of the Abstract (very long!) is not grammatical, and needs rewording.

The sentence mentioned is the following:

"Combined with cutting-edge shell-model and barrier penetration calculations, data taken with the TASI Spec decay station at the Accelerator Laboratory of the University of Jyväskylä, Finland, and the ACTAR TPC device on LISE3 at GANIL, France, we measured

their branching ratios as $b_{p1} = 1.3(1) \%$ and $b_{p2} = 0.025(4) \%$, finally elucidated their nature, 50 years after their discovery."

We have reworded it to:

"Combining data taken with the TASISpec decay station at the Accelerator Laboratory of the University of Jyväskylä, Finland, and the ACTAR TPC device on LISE3 at GANIL, France, we measured their branching ratios as $b_{p1} = 1.3(1) \%$ and $b_{p2} = 0.025(4) \%$. These results were compared to cutting-edge shell-model and barrier penetration calculations. This new description reproduces the order of magnitude of the branching ratios and partial half-lives, despite their extremely small spectroscopic factors."

2. Line 55: the energy of the 19/2- isomer is given as 3174 keV. The same value is shown in Fig.1 with the reference to the NNDC data base (Ref.19). However, the NNDC gives 3197 keV for this state. If the authors use a newer value, the proper reference should be given. Similarly, the half-life of the isomer is given by NNDC as 247(12) ms, and not 245(10) ms, as in line 96 and in Fig.1. It should be made consistent or explained.

The excitation energy 3174.3(10) keV has been determined with the JYFLTRAP Penning trap in A. Kankainen et al., Physical Review C 82, 034311 (2010). This work was not included in Ref. 19 (ENSDF for 53Co), which is based on Huo Junde NDS 110,2689 (2009) with a literature cut-off date 31-Mar-2007. We thank the referee for pointing this out. We have updated reference 19 to refer precisely to the nuclear data sheets of 53Co [Huo Junde Nuclear Data Sheets 110, 2689 (2009)] and added a new one for the 53Co p decay [Yang Dong, Huo Junde NDS 128, 185 (2015)]. A reference to the JYFLTRAP measurement has been added in the figure caption of Figure. 1: "Decay scheme of 53Com based on previous [Junde2009, Dong2015, Kankainen2010] and present results."

The half-life of the isomer was determined as 239(21) ms in this work using ACTAR-TPC. Combined with the literature value, 247(12) ms, from [Huo Junde NDS 110,2689 (2009)], this yields a half-life of 245(10) ms. A sentence has been added to the figure caption of Fig. 1: "The half-life of the isomer is based on the ACTAR-TPC measurement and literature [Huo Junde NDS 110, 2689 (2009)]."

3. Line 60: for the list of proton emitters the Ref. 5 is given. A somewhat newer version was given by Pfutzner et al RMP 84 (2012) 567. The most up-to-date source is the Berkeley data base: <https://nucleardata.berkeley.edu/research/betap.html>

We thank the referee for the attention to details. The sentence is intended merely to note that there are many more proton emitters. The reference list has been updated to include both suggested ones. The text has been adapted and now it reads "over 60 protons emitters" instead of "about 45 proton emitters" as well.

4. Line 77: the sentence is not clear, something is missing.

The full sentence in question is the following:

“The determination of the proton decay width offers a powerful means to characterize the isomeric state because of its sensitivity to the fine details of the wave function, the latter of interest in $^{53}\text{Co}^m$ due to its peculiar structure (full alignment in the angular momentum) and its proximity to doubly-magic ^{56}Ni .”

And to add clarity it has been changed to:

“The determination of the proton decay width offers a powerful means to characterize the isomeric state because of its sensitivity to the fine details of the wave function, of particular interest in $^{53}\text{Co}^m$ due to its peculiar structure (full alignment in the angular momentum) and its proximity to doubly-magic ^{56}Ni .”

4. Line 83: invoking the center-of-mass is confusing and it does not make much sense as decays in question occur at rest. Instead, it should be specified if the proton energy is meant, or the decay energy. By comparing the values in line 88 and in Fig.1, apparently the authors mean the decay energy (Q_p) but then, the Q-symbol should be used instead of $E_{\{p,CM\}}$. So please explain and make it consistent.

The notation $E_{\{p,CM\}}$ is used and understood among some communities but we agree that expressing it as Q_p is clearer. Therefore, throughout the paragraph containing the “Line 83”, the symbol $E_{\{p,CM\}}$ has been changed to Q_p . No other mentions of $E_{\{p,CM\}}$ occur in the text.

5. Line 344: the distinction of protons from alpha particles in the ACTAR TPC could be proven by showing examples of such events, or by providing a reference to such examples.

The distinction between proton and alpha particles is understood from the characterization of the ACTAR TPC using GEANT4 simulations at different gas pressures as described in reference Giovanazzo, J. et al., ACTAR TPC performance with GET electronics, Nucl. Instrum. Methods Phys. Res. A953, 163184 (2020). This reference has been added to the text.

6. In the Summary (from line 384) it would be useful to state what is meant by "a complete understanding" of the decay under study. One could summarize here what is the theoretical picture of this decay and what is the achieved breakthrough, if any.

The wording "complete understanding" has been reformulated as "All decay branches of the isomer have been experimentally measured and a new theoretical description has been proposed that reproduces the order of magnitude of the branching ratios and partial half-lives, despite their extremely small spectroscopic factors."

7. Typographical points:

- in the nuclear literature an isomer is usually denoted by placing the 'm' in superscript between the mass number and the chemical symbol (^{53m}Co). Placing the 'm' on the right side may be confusing when it appears with the charge of an ion ($m+$), like in lines 124, 128;
- the physical units, like MeV, fm, mm, should be written in regular font, and not italic;
- line 295: a mathematical sign was rendered wrongly.

Although we borrowed the notation using the 'm' on the right side of the chemical symbol from the literature itself, in particular the early papers on proton radioactivity by Cerny et al., we understand the confusion raised by the reviewer and consequently the notation has been changed in the text and figures to have the 'm' on the left side of the chemical symbol.

The physical units are now written in regular font.

The mathematical sign in question is "~" and its proper display was ensured. This is also true for other symbols and equations in the text.

Reviewer #2 (Remarks to the Author):

The one-proton radioactivity of the $19/2^-$ isomeric state of Co-53 has been attracted attention by researchers for many years. For the transition from $19/2^-$ isomeric state of Co-53 to $0+$ ground state of Fe-52, its branching ratio was determined as about 1.5% in 1972. Although the one-proton radioactivity of Co-53m was remeasured in the past decades, the branching ratio from $19/2^-$ to the first excitation state ($2+$ state) was not determined accurately because its branching ratio is so small ($<0.006\%$) that it is impossible to observe the proton radioactivity with such a small branching ratio. In this manuscript, the authors measured the branching ratios from $19/2^-$ to $0+$ and $2+$ states by the JYFL-TASISPEC and GANIL ACTAR TPC. The two branching ratios were determined as 1.3(1)% and 0.025(4)%, respectively. Furthermore, the hindered proton-radioactivity with high angular momentum was analyzed by shell models. So this work is interesting and important for understanding the structure and decay properties of Co-53. Before publication on NC, a few minor revisions are advised to be considered by the authors:

1. In paragraph 1 of page 5, the authors expressed "...which yields partial half-lives of 18.8(16) s and 968 (160) s...". However, in the last column and the first line of Table 1, the listed partial half-life is 18.6(16) s. The data must be checked by the authors.

We thank the reviewer for such careful reading. It turns out that both reported half-lives need correction. The new values of 18.8(16) s and 980(162) s are now verified in both the text and the table.

2. In the discussion section, the spectroscopic factor was estimated. However, the details for calculating the spectroscopic factor was not shown. At least, a few equations or formulas that are correlated to the spectroscopic factor should be given.

The following test has been added to the discussion section:

“The spectroscopic factor²⁵ is given by the reduced matrix element

$$S_p = \frac{|\langle \Psi(^{52}\text{Fe})_f J_f | \tilde{a}_{n,\ell,j} | \Psi(^{53}\text{Co})_i J_i \rangle|^2}{(2J_i + 1)}$$

where $\tilde{a}_{n,\ell,j}$ is a single-proton destruction operator, $J_i = 19/2$ and $J_f = 0$ or $J_f = 2$. The spectroscopic factors summed over all final states (f, J_f) gives the orbital occupation number for the orbital (n, ℓ, j) in the initial state (i, J_i).”

[25] Macfarlane, M. H. & French, J. B. Stripping Reactions and the Structure of Light and Intermediate Nuclei. *Rev Mod Phys* **32**, 567 (1960).

Accordingly, the nomenclature C^2S has been dropped in favor of S_p for the spectroscopic factor throughout the text.

3. Has the spectroscopic factor been calculated by other models? What’s the order of magnitude within by other models? Are the spectroscopic factors dependent on different models?

To address this question, we have reworded the paragraph:

“The decay rates are products of large barrier-penetration factors and tiny spectroscopic factors; and those factors have been estimated previously¹¹ and an attempt for a quantitative calculation has been carried out for the spectroscopic factors. The decay width can be factorized into two components”.

To

“The decay rates are products of large barrier-penetration factors and tiny spectroscopic factors; in the original paper of Cerny et al.², a single-particle half-life for $19/2^-$ to 0^+ , $\ell_p=9$, of 60 ns was obtained from a standard barrier penetration calculation and a spectroscopic factor of 1×10^{-6} was estimated by Peker et al.²⁴ in a simple model for the wavefunctions

and residual interaction; none of them provided a satisfactory description. In this work we factorized the decay width into two components”

[24] L. K. Peker, E. I. Volmyansky, V. E. Bunakov and S. G. Ogloblin, Phys. Lett. 36B, 547 (1971).

4. In paragraph 2 of page 6, the proton-decay width $\Gamma_{sp} = WP(\rho)$. What does the “W” stand for?

To add clarity to the discussion section we have expanded the description of the terms by exchanging the following part of the text:

“The proton-decay widths were also calculated with the approximation $\Gamma_{sp} = WP(\rho)^{26}$, where P is the penetration factor obtained from the Coulomb wave function, and $W = 1.155$ fm works for both $\ell_p = 7$ and $\ell_p = 9$. With this, the barrier penetration model gives the same result as the Woods-Saxon scattering calculation over the Γ_{sp} range from 5×10^{-11} MeV to 1×10^{-8} MeV to within one percent. The barrier-penetration model can be extrapolated down to the Q values needed for proton emission of ^{53}Co that have widths of the order of 10-15 MeV.”

by:

“The single-particle proton widths were also calculated from³⁰

$$\Gamma_{sp} = 2\gamma^2 P(\ell, R_c, Q_p),$$

with $\gamma^2 = \frac{\hbar^2 c^2}{2\mu R_c^2}$ and we obtain the Coulomb penetration, P , from Barker³¹. The channel radius, R_c , was chosen to match the decay widths obtained from the Woods-Saxon potential. A value of $R_c = 5.46$ fm works for both $\ell_p = 7$ and $\ell_p = 9$. With this, the barrier penetration model gives the same result as the Woods-Saxon scattering calculation over the Γ_{sp} range of from 5×10^{-11} MeV to 1×10^{-8} MeV to within about one percent. The barrier-penetration model can be extrapolated down to the Q values needed for proton emission of ^{53m}Co that have single-particle decay widths of the order of 10^{-15} MeV.”

Reviewer #3 (Remarks to the Author):

In the proton radioactivity a quasi-bound proton tunnels through the Coulomb and centrifugal barrier, leaving the initial state and populating states in the nucleus with one less proton. Naturally, this phenomenon occurs in proton rich nuclei at or near the proton drip line. The present paper deals with the proton decay of the $^{53}\text{Co}^m$ isomer, in which the proton radioactivity was first observed already 50 years ago. Now it is possible to study the process quantitatively and fully, and determine even the branching ratio of the very weakly populated $2+$ state in the final nucleus ^{52}Fe . This is in some sense a continuation of Ref. [7] published in Nature Communications in

2021 by a subset of the coauthors of the present paper. Ref. [7] dealt with the proton decay of $^{54}\text{Ni}^m$ isomer, i.e. the nucleus with one more proton than here. Analogous devices, e.g. the ACTAR TPC detector at GANIL are used in both papers; the theoretical analyses are also similar in both papers. However, the present paper not only uses in addition the JYLTRAP Penning trap at Jyvaskyla, but deals with much weaker, thus more difficult to observe, proton decay branches. I believe that the Nature Communications journal is the right place to publish the present results.

My minor suggestion is to add, either on a separate line or in the text, the formula connection the partial half-life with the width ($T_{1/2,sp} = \hbar \ln(2) / \Gamma_{sp}$) as well as the partial half-life with the total one ($T_{1/2} = T_{1/2,sp} / C_{2S}$). That would make the understanding of Table 1 easier to an educated non-expert. I agree with the authors that the agreement between the measured and calculated branching ratios is remarkable given the approximations used. Not only is the model space truncated, with only one nucleon allowed to move out of the $f_{7/2}$ orbit, but also the barrier penetration model is extrapolated by many orders of magnitude from the $\Gamma_{sp} \sim 10^{-11}$ MeV to about 10^{-16} MeV (the text mentioned 10^{-15} , I believe the entries in Table 1 correspond rather to about 10^{-16}). The theoretical uncertainty might be actually larger than the factor of three mentioned. I hesitate what to suggest but maybe instead of 'close to a factor of three' one could say 'at least a factor of three'. I leave it to the authors whether they choose to follow my minor suggestions or not. I recommend the publication in Nature Communications.

The following has been done to address the minor suggestions made by the Reviewer.

New formulas contextualizing the partial half-life with the width and the total half-life. This was changed:

“The results for $T_{1/2,sp}$ are given in Table 1” by “The results for $T_{1/2,sp} = \hbar \ln(2) / \Gamma_{sp}$ are given in Table 1”. The equation $\Gamma = S_p \cdot \Gamma_{sp}$ is now included in the text as well. We believe these changes, in connection with the ones made in connections to the comments from Reviewer 2, enhance the readability of the section for non-expert readers.

The values for the barrier penetration used to obtain the values in Table 1 have been carefully checked. We made a transcription error for the (0,9,19/2) spectroscopic factor. Instead of 0.18×10^{-6} it now reads 0.062×10^{-6} and the theory half-life changed from 19 s to 55 s

The phrasing 'close to a factor of three' referring to the theoretical uncertainties has been addressed by changing the original paragraph:

“With this change the spectroscopic factor for $\ell_p = 7$ is increased by about a factor of two, and the spectroscopic factor for $\ell_p = 9$ is decreased by about a factor of three. We conclude therefore that the uncertainty in the calculations is close to a factor of three. The calculated results agree with experiment within a factor of two. Given the exceptionally large hindrance factors, this is in remarkably good agreement.”

to

“With this change the spectroscopic factor for $\ell_p = 7$ is increased by about a factor of two, and the spectroscopic factor for $\ell_p = 9$ is decreased by about a factor of two. The calculated results are of the same order of magnitude as the experimental values; a maximum discrepancy of a factor of two to four was found. Given the exceptionally large hindrance factors, this is a remarkably good agreement.”

In connection with this rewording, the caption for Table 1 was simplified.

REVIEWERS' COMMENTS

Reviewer #1 (Remarks to the Author):

The Authors did answer all my questions and introduced all suggested corrections. The paper is clearly improved. I do not have any further critical comments. The presented result is very nice and important, but it is not a breakthrough of any kind. That is why I am not convinced that the Nature Communication is the proper place for its publication. But I do not mind being outvoted by other referees and editors.

Reviewer #2 (Remarks to the Author):

The authors have revised the manuscript as my suggestions and comments. In my opinion, it has met the published requirement of Nature Communications. So I recommend its publication on Nature Communications.

Replies to the reviewers

Reviewer #1 (Remarks to the Author):

The Authors did answer all my questions and introduced all suggested corrections. The paper is clearly improved. I do not have any further critical comments. The presented result is very nice and important, but it is not a breakthrough of any kind. That is why I am not convinced that the Nature Communication is the proper place for its publication. But I do not mind being outvoted by other referees and editors.

We thank the reviewer for the initial careful reading.

Reviewer #2 (Remarks to the Author):

The authors have revised the manuscript as my suggestions and comments. In my opinion, it has met the published requirement of Nature Communications. So I recommend its publication on Nature Communications.

We thank the reviewer for the initial careful reading.

Reviewer #3 (Remarks to the Author):

<No new remarks from Reviewer #3 were communicated to us>

We thank the reviewer for the initial careful reading.